# The effect of support surface and footwear condition on postural sway and lower limb muscle action of the older women

**Meizhen Huang**[1,2], **Kit-lun Yick**[3]*, **Sun-pui Ng**[4], **Joanne Yip**[3], **Roy Tsz-hei Cheung**[1,5]

**1** Department of Rehabilitation Sciences, Gait & Motion Analysis Lab, The Hong Kong Polytechnic University, Hung Hom, Hong Kong, **2** Department of Physical Therapy and Rehabilitation Science, University of Maryland School of Medicine, Baltimore, Maryland, United States of America, **3** Institute of Textiles and Clothing, The Hong Kong Polytechnic University, Hung Hom, Hong Kong, **4** College of Professional and Continuing Education, The Hong Kong Polytechnic University, Hong Kong, **5** School of Health Sciences, Western Sydney University, Campbelltown, New South Wales, Australia

* tcyick@polyu.edu.hk

## Abstract

### Background

Diminished somatosensory function is a critical age-related change which is related to postural instability in the older population. Footwear is a cost-effective way to modulate the postural stability by altering sensorimotor inputs via mechanoreceptors on the plantar surface of the feet. Compared to insoles with indentions in the entire surface, we innovatively developed a textured insole with site-specific nodulous protrudous. This study thus aimed to investigate the immediate effect of the nodulous insole and supporting surface condition on static postural stability and lower limb muscle activation for healthy older women.

### Methods

This is a single-session study with repeated measurements. Twenty-three healthy older women stood on the firm (i.e., concrete floor) and foam surfaces with their eyes open in the three footwear conditions, namely barefoot, plain shoes and shoes with an innovative textured insole, for 30 seconds. Static postural sway and muscle activation of biceps femoris (BF), vastus lateralis (VL), tibialis anterior (TA), and lateral gastrocnemius (LG) of the dominant leg were measured during each testing condition.

### Results

Compared to a firm surface, standing on the foam could significantly increase the body sway and lower limb muscle activation ($p<0.05$). When standing on the foam, compared to barefoot, wearing footwear significantly decreased the VL and TA muscle activation and minimize the postural sway in medial-lateral and anterior-posterior direction, while the influence is larger for the shoes with nodulous insloe compared to the plain shoes. No significant differences between the footwear conditions for static stability and muscle activation were observed on firm surface condition.

**Data Availability Statement:** All relevant data are within the manuscript and its Supporting Information files.

**Funding:** The work is supported by funding from the Innovation and Technology Fund (ITF) (ITS/359/14) and the Innovation and Technology Commission (AiDesign Lab Project).

**Competing interests:** The authors have declared that no competing interests exist.

## Conclusions

For older women, footwear could improve the postural stability in the unstable surface, particularly the footwear with nodulous insole, with the underlying mechanism as enhancing the mechanoreceptors on the plantar surface of the feet.

## Background

Balance is a major concern for older people [1–4]. The decline in postural stability in older adults increases the risk of falls that ultimately elevates the morbidity, mortality, and cost of health care services [1–4]. To maintain the normal static bipedal stance, individuals primarily depend on the proprioceptive and cutaneous input [5, 6]. However, diminished somatosensory function including lower plantar cutaneous sensitivity is a critical age-related change, which may increase the body sway for the older population and increase the fall risk [5–11]. And the fall risk and fall-related injury are higher in women than men of the similar age [1–3]. For community-dwelling women, a majority of fall happen at home [4] due the environmental factors such the uneven floor [2–4]. Indeed, unreliable sensory stimulus from the unstable surface would further deteriorate the postural control [6, 9, 12, 13]. While the unstable surfaces are commonly encountered in various daily life settings including thick carpets, it is imperative to develop practical strategy to enhance postural stability for the older adults.

Footwear is a cost-effective way to modulate the postural stability by augmenting plantar cutaneous sensations [14–17]. A simple approach is to facilitate the sensation is by applying indentations on the footwear insole to stimulate the mechanoreceptors [14–16]. For instance, rigid textured insole with dimpled and grid surfaces was reported to reduce the postural sway for the older people when compared to the barefoot condition [18–20] or shoes without textured insole [20]. Also, hard insoles can effectively enhance the postural stability when visual input was absent for the older adults [21] and insoles with soft texture was unanimously suggested to be detrimental for postural control for the older people [14]. However, as reported by previous research using the rigid insole with indentions on entire surface, "most participants anecdotally reported that the harder insoles were uncomfortable to stand on for an extended period of time" [18]. Indeed, it has been recognized that flexible and relatively soft soles are preferable to hard soled shoes [22]. Thus, a dilemma exists that hard insole with indentions could enhance postural control but provide discomfort for the users, which may limit its application.

To address the problem, we have developed an innovative textured insole by applying silicone nodules in the metatarsal, heel, and the foot arch area. The underlying mechanism including (1) mechanoreceptors mainly locate in the metatarsal-tarsal and heel region of foot sole [23]; (2) investigation regarding electrical site-specific stimulation to these areas has implied consequential improvement of balance [24]; (3) the raised nodules create a boundary of the plantar surface from the metatarsal heads to the heel that has been suggested to facilitate postural stabilizing reactions evoked by unpredictable postural perturbation [25]. We proposed that this design would largely maintain users' comfort while facilities the plantar somatosensory input, particularly in the unstable surface which is common but risk for the older women.

This study, therefore, aimed to investigate the immediate effect of the novel nodulous textured insole and supporting surface condition on static postural stability and lower limb muscle activation for older women. We hypothesized that compared to plain shoes and barefoot,

novel nodulous textured insole could significantly enhance the postural stability and minimize the muscle lower limb muscle activation, particularly in the foam surface, for older women.

## Methods

This study was approved by the Human Subjects Ethics Sub-committee of The Hong Kong Polytechnic University (Reference Number: HSEARS20150806001). All the participants provided written informed consent before data collection.

### Participants

Independent community-dwelling older women were recruited through convenient sampling. The inclusion criteria were: (1) aged more than 60; (2) able to follow simple verbal instructions, (3) no history of foot injuries during the past two years. The exclusion criteria included: (1) having any neurological conditions and musculoskeletal problems that might affect balance; (2) having contract or glaucoma; (3) people with impaired tactile foot sensitivity. The study was in accordance with the latest revision of the Declaration of Helsinki [26].

### Study procedures

This was a single-session study with repeated measurements. Participants were recruited from February to March in 2016. We first interviewed participants to obtain their demographic information, following by foot sensation test and static balance test in different footwear conditions. The dominant limb was determined by a ball kicking task [27].

### Foot sensation test

Tactile foot sensitivity was evaluated using Semmes Weinstein Monofilament test (North Coast Medical Inc, California, United States) which was widely used valid tactile sensory [28]. Monofilaments, starting with 1.65 mm, were randomly applied to test regions including first, second, third toes, first, third, and fifth metatarsal heads, and medial and lateral arches of the midfoot. Participants were asked to close eyes and indicate whether stimulations were perceived. A maximum of three stimulations per monofilament was applied on each region. The inability to detect 5.07 mm/10 g monofilament at first toes, first and third metatarsal heads was deemed as impaired tactile foot sensitivity [29].

### Testing conditions

Each participant then underwent three different footwear conditions, namely barefoot, plain shoes made of soft terry textiles (13F302, Yeshunag, Zhejiang, China) (Fig 1a), and nodulous insole shoes with medial arch support and silicone protrusions at the metatarsal heads and lateral heel (KE-1300T, Shin-Etsu Chemical Co., Ltd.) (Fig 1b), in a randomized sequence. The barefoot condition was simulated by placing pressure-sensing insoles onto the bottom of the feet, and then secured by standard cotton socks that are 1.4 mm in thickness.

Participants were instructed to maintain quiet standing on firm (i.e., concrete floor) and foam surfaces (46.5 cm (L) × 46.5 cm (W) × 4.5 cm (H); 0.0764 g/cm3, StimUp® Balance Pad, Will Medical, Tokyo, Japan) with their eyes open in the three footwear conditions. Thus, there were six experimental conditions which were carried out in a randomized order. The standing posture was standardized according to the method used in a previous study [30]. In brief, all the participants' feet were 17 cm apart from the heel center, with the foot progression angle at 14˚. They were asked to stand still with their hands by the sides and looking at a stationary visual target (i.e., a red spot of 2 cm in diameter) placed at eye level and 3-m in front of the

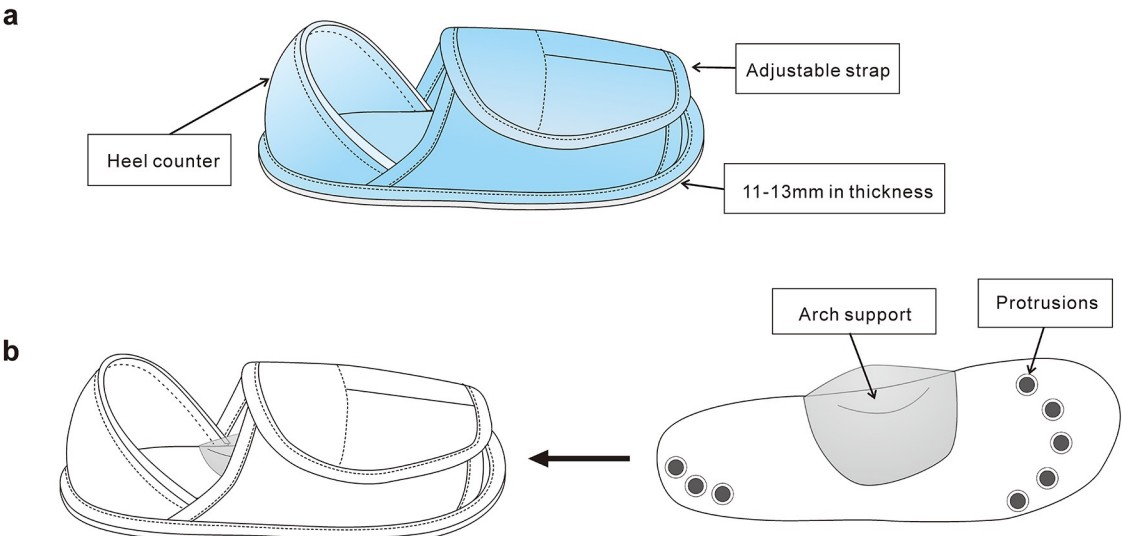

**Fig 1. Test shoe models.** (a) The plain shoes are open-toe design with secured heel counter and an adjustable dorsal forefoot strap. The sole thickness is uniform across the plantar; (b) The nodulous shoes is the same plain shoes with raised nodulous and medial arch support.

participants and put equal body weight on each foot. In each condition, participants were tested thrice for 30 seconds and a rest period of one minute was allowed between each condition [30]. Data during the middle 20 seconds were used for analyses.

## Measurement of static postural stability

Static postural sway was measured as the displacement of center of pressure (COP) using in-shoe pressure measurement system (Pedar®, Novel GmbH, Munich, Germany) with sampling frequency at 50 Hz under the dominant foot. The Pedar® insole sensor, which was 2 mm in thickness and fitted in accordance with foot size, was placed underneath the foot. Plantar pressure by Pedar® insole sensor had been reported with good validity and reliability [31, 32] and provided a valid approximation of COP [33, 34]. Referring to previous studies [18, 30], the COP parameters included the range of anterior-posterior (AP) and medial–lateral (ML) COP displacement, path length (PL) and the 95% confidence elliptical area (C95 area).

## Measurement of electromyography

The surface EMG signal was captured using an 8-channel wireless EMG system (Clinical DTS, Noraxon USA Inc., AZ, USA) with 16-bit resolution and common-mode rejection ratio > 100 dB. After proper skin preparation, four circular Ag/AgCl bipolar electrodes (electrode diameter: 10 mm, inter-electrode distance: 22 mm) were placed on the biceps femoris (BF), vastus lateralis (VL), tibialis anterior (TA), and lateral gastrocnemius (LG) of the dominant leg according to the recommendations of Surface EMG for Non-Invasive Assessment of Muscles [35]. The raw signals were pre-amplified 1,000 times and sampled at 3,000 Hz with 500 Hz low-pass filter. Prior to postural stability test, maximum voluntary contractions (MVC) for each muscle was collected for 5 s using manual resistance [36] and repeated three times with five minutes rest period in-between each MVC test (S1 Appendix).

## Data extraction and processing

The AP and ML COP displacement, COP path length and the 95% confidence elliptical area (C95 area) parameters were calculated by the inbuilt pressure measurement system (Pedar$^{®}$/expert Software, Novel GmbH, Munich, Germany).

The EMG signals were processed using MyoResearch 3 (Noraxon USA Inc, AZ, USA). All EMG data were filtered using a first-order high-pass Butterworth filter at 10 Hz with 8–10% cutoff. The data were then rectified, and the root-mean-square ($EMG_{rms}$) was calculated in 100 ms windows during the 20-second time-window of the static standing trials. For each muscle, the highest $EMG_{rms}$ portion of 100 ms duration from the three MVC trails for each muscle was extracted and averaged, which was then used for normalization of the $EMG_{rms}$ value in each testing condition (i.e., $EMG_{mvc\%}$) [37].

## Statistical analysis

Statistical analyses were conducted using SPSS (version 22, IBM, Armonk, NY). We used Shapiro-Wilk test and probability-probability plot to check the normality of dependent variables. Only $EMG_{mvc\%}$ was positively skewed and log transformation was thus applied to the data to obtain normally distributed responses [38].

Two-way repeated-measures ANOVA (within-subject factors: three footwear conditions, two supporting surface conditions) was applied to each static stability and EMG variables. Greenhouse-Geisser epsilon adjustment was used when the sphericity assumption was violated. Post-hoc analysis using paired t-test with Bonferroni adjustment was performed if any overall significant results were identified. The effect size was expressed as partial eta squared ($\eta_p2$) [39]. A significance level of $p \leq 0.05$ was set for two-way repeated ANOVA models.

# Results

Twenty-three older women (mean age: 65.1±3.3 years) completed all the tests. The demographics are summarized in Table 1. Participants exhibited tactile foot sensitivity level within the normal threshold [29].

Significant footwear condition × supporting surface condition was observed for VL $EMG_{mvc\%}$ (p<0.001, $\eta_p2 = 0.308$), TA $EMG_{mvc\%}$ (p = 0.002, $\eta_p2 = 0.250$), AP displacement (p = 0.002, $\eta_p2 = 0.249$), COP path length (p = 0.005, $\eta_p2 = 0.211$), COP velocity (p = 0.004, $\eta_p2 = 0.251$), 95% confidence elliptical area (p = 0.013, $\eta_p2 = 0.179$), which indicates significant interactions.

## Effect of supporting surface condition

Significant effects of supporting surface were observed for all the EMG variables and postural stability (p<0.001, $\eta_p2 = 0.150–0.782$) except total path length and COP velocity. Significant larger postural sway parameters and $EMG_{mvc\%}$ was observed when standing on the foam.

**Table 1. Demographic data of the participants.**

| Demographic data | Range | Mean ± SD |
|---|---|---|
| Age (years) | 60.0–73.0 | 65.1 ± 3.3 |
| Tactile foot sensitivity (mm) | 3.61–4.56 | 3.95±0.36 |
| Body height (m) | 1.42–1.66 | 1.54 ±0.55 |
| Body mass (kg) | 38.0–65.0 | 51.7±6.3 |
| BMI (kg/m2) | 15.2–32.2 | 22.2±3.4 |
| Foot size (Euro) | 35–40 | 37.4±1.1 |
| Dominant leg | Right (n = 23); Left (n = 0) | |

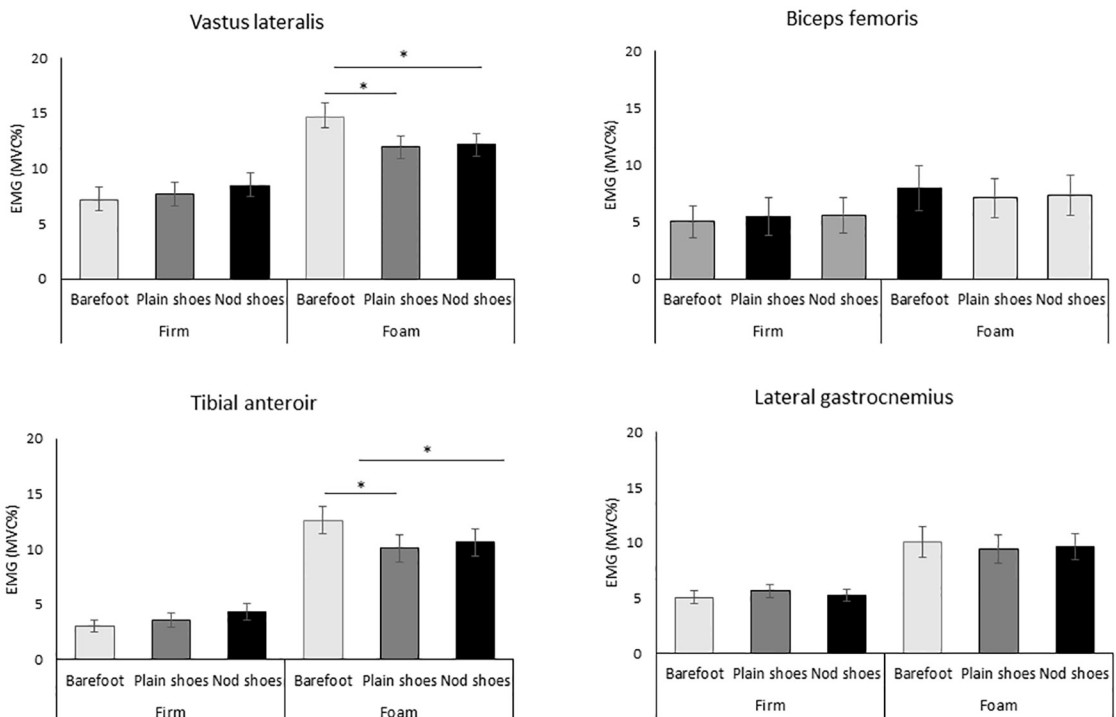

**Fig 2. Lower limb muscle activity.** Nod shoes: nodulous shoes. * Significant difference of pair-t test, p<0.05 with Bonferroni adjustment.

### Effect of footwear conditions

A significant effect of the footwear was observed in foam but not the firm surface for VL $EMG_{mvc\%}$ (p<0.001, $\eta_p2 = 0.723$), TA $EMG_{mvc\%}$ (p = 0.047, $\eta_p2 = 0.101$), AP displacement (p = 0.03, $\eta_p2 = 0.267$), ML displacement (p = 0.03, $\eta_p2 = 0.271$), C95 area (p = 0.002, $\eta_p2 = 0.254$). Post-hoc analysis showed that the smallest value was observed with nodulous shoes, and highest value was in barefoot (p<0.05) (Figs 2 & 3).

## Discussion

Our finding demonstrated the nodulous textured insole would enhance the postural in the foam condition for the older women. Specifically, (1) when compared to firm surface, standing on the foam could significantly increase body sway and lower limb muscle activation, regardless of the footwear conditions; (2) when standing on the foam, comparing to barefoot, wearing footwear could immediately decrease the lower limb muscle activation and minimize the postural sway in ML and AP direction, while the influence was larger for the nodulous shoes compared to the plain shoes; (3) No significant differences between the footwear conditions for static stability and muscle activation was observed on firm surface condition.

In agreement with the previous studies, standing on foam surface increases the postural sway [18, 19]. Meanwhile, the effect of footwear condition on the postural sway was only observed on the foam surface, not the firm surface [18, 19]. When standing on a foam surface, the accuracy of the proprioception input was affected [6, 13], thus more demanding for sensory input from the mechanoreceptors.

As expected, the nodulous insole significantly enhanced the postural stability when standing on the foam surface, demonstrating by the reduction in AP and ML sway, and reduced

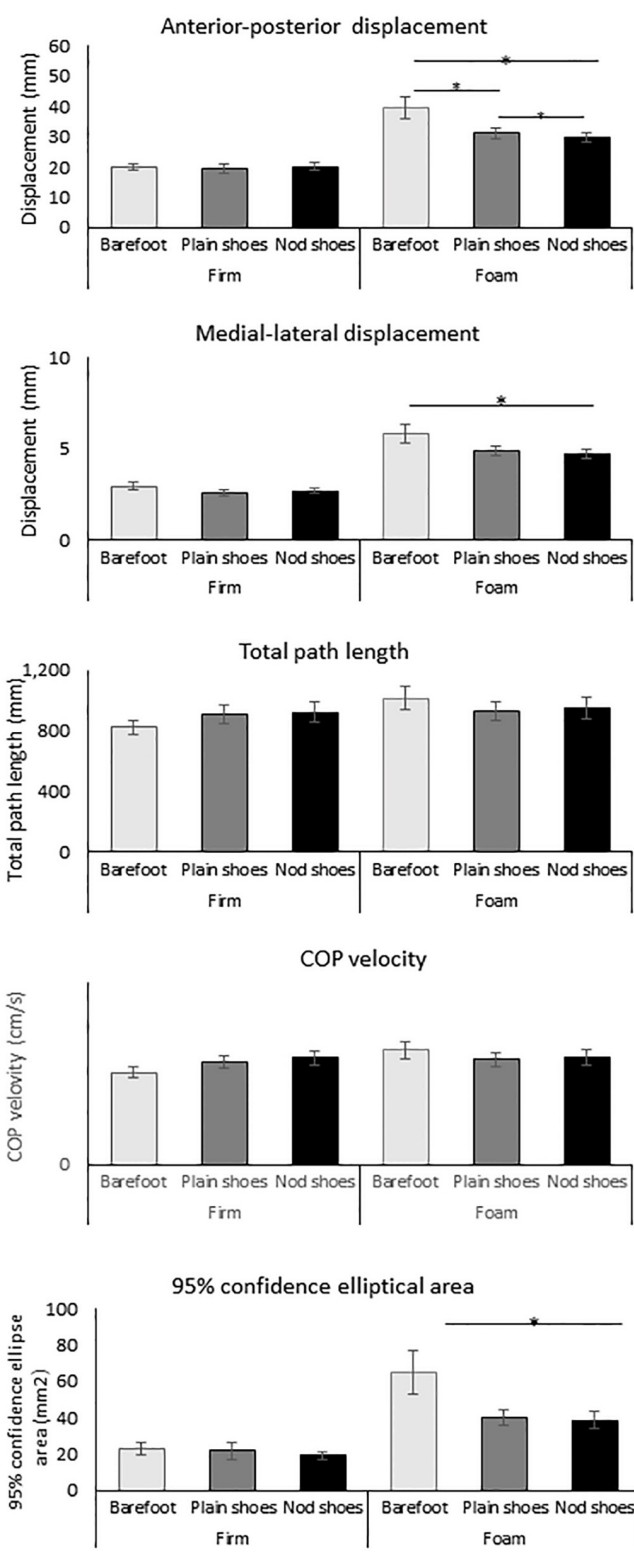

**Fig 3. Postural sway.** Nod shoes: nodulous shoes. * Significant difference of pair-t test, p<0.05 with Bonferroni adjustment.

C95 area; while plain shoes only show reduced AP sway on the foam surface (Fig 3). These findings were also comparable with previous studies applying hard insole with dimples covering the entire insole [18–20]. ML sway is an important clinical parameter because ML sway has been shown to increase with age more than AP [40] and be greater in fallers than age-matched non-fallers [9]. Several possible mechanisms regarding the static posttrial control by insole have been suggested. First, the nodulous surface may produce higher plantar pressure at the elevated parts of the textured sole, providing stronger sensory stimulation to the mechanoreceptors [18–20, 41]. We also specifically placed the nodulous in the metatarsal-tarsal and heel region, where is the mechanoreceptors predominately locate [23]. Second, footwear is also suggested as a filter for the pressure input to the sole [42], which may remove the disturbance information during standing on the foam. Therefore, we observed a trend that wearing plain insole shoes may also improve the postural stability in our study. Third, footwear may optimize the biomechanical alignment of lower limbs, thus promoting the efficiency of postural control [16, 42]. We created arch support for the nodulous insole (Fig 1). However, no difference in muscle activation was shown in the between the nodulous and plain shoes, though better postural control was observed in the condition of the nodulous shoes. This may imply the influence due to biomechanical alignment change would be trivial. Collectively, nodulous insole shoes could promote the static postural control possibly due to enhancement of mechanoreceptors in the sole.

One interesting finding of the present study is that footwear could significantly decrease muscle activity for the vastus lateralis and tibial anterior muscle but not other tested muscles. Standing on an unstable supporting surface largely challenge the stability of knee joint [43] and subtalar joint [44]. Thus, knee extensors (e.g., VL) would be greatly activated to main the knee stability. Tibial anterior, the major foot inverter muscles, activate with other extrinsic and intrinsic foot muscles to maintain proper foot alignment [44]. Thus, in consistent with previous findings, standing on the foam significantly increase the TA muscle activation [43, 45].

This study acknowledges a few limitations. First, only older women were involved in this study. The previous study suggested that the gender difference might exist in balance control and somatosensory function for the healthy elderly [46], thus recruiting the same gender in this study could minimize the potential confounding factors. However, future studies should examine both male and female participants, and also other individuals with neurological conditions [41]. Second, for the barefoot condition, the participant stood on the pressure sensor. However, the insole sensor is flexible with 2-mm thickness, which could largely preserve the barefoot condition. Third, only muscle activation and COP changes in the dominant foot were measured. In this study, we standardize the feet position (i.e., feet were 17 cm apart from the heel center, with the foot progression angle at 14˚), and the participants were instructed to maintain equal body weight between two feet. Thus, the measurement in the dominant side could represent the performance of the lower limb. More, the measures of the insole pressure sensor are reported comparable to the traditional force plate [46]. Fourth, the foam surface was used in this study. Although standing on foam does not have much external validity as it is not representative of real life situation, e.g. standing on soil or other unstable surfaces, this condition provides more information on the effect of footwear to balance control as the somatosensory information is challenged.

## Conclusion

For older women, footwear could improve the postural stability in the unstable surface, particularly the footwear with nodulous insole, with the underlying mechanism as enhancing the mechanoreceptors on the plantar surface of the feet.

## Supporting information

**S1 Appendix. Testing maneuver of maximum voluntary contractions.**
(DOCX)

**S2 Appendix. List of abbreviations.**
(DOCX)

**S1 Data.**
(XLSX)

**S2 Data.**
(XLSX)

**S3 Data.**
(XLSX)

**S1 File.**
(DOCX)

**S2 File.**
(CSV)

## Author Contributions

**Conceptualization:** Meizhen Huang, Sun-pui Ng.

**Data curation:** Meizhen Huang, Kit-lun Yick.

**Funding acquisition:** Kit-lun Yick, Sun-pui Ng, Joanne Yip, Roy Tsz-hei Cheung.

**Methodology:** Meizhen Huang, Roy Tsz-hei Cheung.

**Project administration:** Kit-lun Yick.

**Resources:** Kit-lun Yick, Roy Tsz-hei Cheung.

**Software:** Kit-lun Yick.

**Supervision:** Kit-lun Yick.

**Validation:** Meizhen Huang.

**Writing – original draft:** Meizhen Huang.

**Writing – review & editing:** Kit-lun Yick, Sun-pui Ng, Joanne Yip, Roy Tsz-hei Cheung.

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
