## [Decision Letter · Decision Letter 0]

20 Mar 2020

PONE-D-20-00586

The effect of support surface and footwear condition on postural sway and lower limb muscle action of the old people

PLOS ONE

Dear Dr. Yick,

Thank you for submitting your manuscript to PLOS ONE. After careful consideration, we feel that it has merit but does not fully meet PLOS ONE’s publication criteria as it currently stands. Therefore, we invite you to submit a revised version of the manuscript that addresses the points raised during the review process.

As you can see, the reviewer 2 ask you to introduce some references. Please add it only if, after reviewing the papers, they are related with the rationale. You do not have to if they are not in line with the writting. Please, address all the editor' comments (below).

We would appreciate receiving your revised manuscript by Apr 27 2020 11:59PM. To enhance the reproducibility of your results, we recommend that if applicable you deposit your laboratory protocols in protocols.io, where a protocol can be assigned its own identifier (DOI) such that it can be cited independently in the future. For instructions see: http://journals.plos.org/plosone/s/submission-guidelines#loc-laboratory-protocols

We look forward to receiving your revised manuscript.

Kind regards,

Pilar Serra-Añó

Academic Editor

PLOS ONE

Journal Requirements:

2. Thank you for stating the following in the Acknowledgments Section of your manuscript: "The work is supported by funding from the Innovation and Technology Fund (ITF) (ITS/359/14) and the Innovation and Technology Commission (AiDesign Lab Project)."

ABSTRACT

In the abstract, the results of the condition without foam must be provided.

Why do the results of correlations appear in the abstract and are not explained as objectives?

This is also the case in the introduction. Correlations appear as hypotheses but are omitted as objectives. Nor is there any rationale in the introduction that explains the need to evaluate these relationships between variables. Please add it or remove the correlations analyses.

The conclusions should also be specific as to whether you are comparing with firm ground or with foam.

Add keywords.

INTRODUCTION

After conducting a general review of the article it is observed that the type of sole has no effect on the firm ground. It only has an effect when tested on an unstable surface.

In this respect, the only thing the authors contribute is "As the healthy participants could achieve their best performance on a firm surface [7, 9], it may not be sufficiently discriminating to detect the change of postural sway across footwear conditions. Indeed, textured insole was reported to reduce the postural sway when standing on foam, when compared to the barefoot condition [9, 10].

The objective of the study should not be to force the support condition to observe differences (fishing), in postural sway, as stated in the paragraph, but rather to determine what is the need to evaluate on unstable surfaces (are they common in the population?) and from there see if differences exist or not according to the type of sole. The approach of the study should change so that the objectives of the study make sense and serve to contribute to the field of footwear design.

Methods

Please review methodological guidelines, such as CONSORT for example, to add the missing methodological aspects that allow replicability of the data. For example, you should add recruitment time, reliability of measurement tools (StimUp® Balance 5 99 Pad, Will Medical, Tokyo, Japan; Pedar®, Novel GmbH, Munich, Germany), etc.

Statistics

Only EMGmvc% was positively skewed? Indicate, if it is true, that the others met the assumption of normality. The statistics used for the main study are very well designed. However, here in this section, there is also no reference to correlations of muscle activation with postural sway.

Results

Change the + symbols to parentheses.

The results do not show the correlations that are in the abstract either.

Change "indicates positive interaction effects" by "significant interaction"

Please, add the legends to the figures.

Discussion

More information on the mechanisms by which alterations in postural control and muscle activation occur is lacking in the discussion. As well as discussing  the differences between what happens on firm ground and on unstable surfaces.

It should be added, in limitations, that the use of foam, as a form of unstable soil does not have much external validity as it is not representative of real life. It would be advisable to slightly modify what the authors have put forward.

Reviewers' comments:

Reviewer's Responses to Questions

**Comments to the Author**

1. Is the manuscript technically sound, and do the data support the conclusions?

Reviewer #1: Yes

Reviewer #2: Yes

2. Has the statistical analysis been performed appropriately and rigorously? 

Reviewer #1: No

Reviewer #2: Yes

3. Have the authors made all data underlying the findings in their manuscript fully available?

Reviewer #1: Yes

Reviewer #2: Yes

4. Is the manuscript presented in an intelligible fashion and written in standard English?

Reviewer #1: Yes

Reviewer #2: Yes

5. Review Comments to the Author

Reviewer #1: This paper examined the footwear insole texture and supporting surface conditions on static posrtural stability and lower limb muscle activation for health older people. While this is a potential very valuable study there is some work needed to get it to the standard for this journal. There were some typographical errors I have included some in specific comments. Please refer to the specific comments below:

1. The effect of support surface and footwear condition on postural sway and lower limb muscle action of the old people

It is suggested that the topic should be change to "The effect of support surface and footwear condition on postural sway and lower limb muscle action of the old women' because only women participants involved in this study.

2. Start with capital letter for 'compared' in the Results abstract section.

3. Start with capital letter for 'for' in the Conclusions abstract section.

4. Change 'old people' to 'old women' in the Background section.

5. What correlation methods/analysis used for your study? There were correlation results in the Abstract but the information was not available in the Results section. There were also no Correlation Analysis information found in the Data Analysis section. The correlation results also were not discussed in the Discussions section. Please revise this issues accordingly.

6. 'pedar' should start with capital letter..i.e. line 130...Please check and revise accordingly.

7. It is good to state the minimum age of your participants, i.e. how many participants are age from 60-65 years old. In the discussion section, you mentioned 'elderly' and by definition, elderly refers to person who are age 65 years old and above...but in Table 1, the age minimum age range of your participants are 60 years old..please revise according regarding the use of 'elderly' and older population/older women.

8. To our knowledge, this is the first study that investigates the footwear condition and supporting surface conditions on static postural stability and lower limb muscle activation for healthy old women.

As to our knowledge, no studies have investigated the effect of footwear insole condition on both static postural stability and lower limb muscle activity for older people.

Please standardized the term used for helthy old momen and older people. What is the main participants involved in this study?

9. As to our knowledge, no studies have investigated the effect of footwear insole condition on both static postural stability and lower limb muscle activity for older people.

The authors should provide a detailed argument before coming to this statement. The information in the same paragraph were only on textured insoles, spike insoles that was inserted to the participants shoes during the testing session. There should be an intensive information on how does the footwear used in this were different from the previous study. Is it a readymade footwear that is available in the market?

10. How do you make sure that all participants have the minimum foot sensitivity level before they were accepted in your study? Do you conducted any test on foot sensitivity?

Reviewer #2: I am grateful for the possibility to revise this research study.

The effect of support surface and footwear condition on postural sway and lower limb

muscle action of the old people is a trend topic in the current research literature and may be a main focus of interest for readers.

Results of the abstract need to reflect the findings with respect to both groups and the lack of significant differences of balance, and also you need reflect the meaning of AP and ML because these abbreviates are not reflected clearly

Introduction may be improved adding new information in order to provide an adequate state-of-the-art including some references. I suggest to include this references include in the atteched to complet this requeriment

Lines 55-68

Rodríguez-Sanz D, Tovaruela-Carrión N, López-López D, Palomo-López P, Romero-Morales C, Navarro-Flores E, et al. Foot disorders in the elderly: A mini-review. Disease-a-Month. 2018 Mar;64(3):64–91.

Calvo-Lobo C, García AR, Iglesias MEL, López-López D, Rodríguez-Sanz D, Romero-Morales C, et al. The relationship between shoe fitting and foot health of persons with down syndrome: A case control study. Int J Environ Res Public Health. 2018 May 14;15(5).

López-López D, Marañon-Medina J, Losa-Iglesias ME, Calvo-Lobo C, Rodríguez-Sanz D, Palomo-López P, et al. The influence of heel height related on quality of life on the foot in a sample of women. Rev Assoc Med Bras. 2018 Apr;64(4):324–9.

Roca-Dols A, Elena Losa-Iglesias M, Sánchez-Gómez R, Becerro-de-Bengoa-Vallejo R, López-López D, Palomo-López P, et al. Electromyography activity of triceps surae and tibialis anterior muscles related to various sports shoes. J Mech Behav Biomed Mater. 2018/06/05. 2018 Oct;86:158–71.

Roca-Dols A, Losa-Iglesias ME, Sánchez-Gómez R, López-López D, Becerro-de-Bengoa-Vallejo R, Calvo-Lobo C. Electromyography comparison of the effects of various footwear in the activity patterns of the peroneus longus and brevis muscles. J Mech Behav Biomed Mater. 2018/03/13. 2018 Jun;82:126–32.

López-López D, Expósito-Casabella Y, Losa-Iglesias M, Bengoa-Vallejo RB de, Saleta-Canosa JL, Alonso-Tajes F. Impact of shoe size in a sample of elderly individuals. Rev Assoc Med Bras. 2016 Nov;62(8):789–94.

López López D, Losa Iglesias ME, Becerro de Bengoa Vallejo R, Palomo López P, Morales Ponce Á, Soriano Medrano A, et al. Optimal choice of footwear in the elderly population. Geriatr Nurs. 2015/08/12. 2015;36(6):458–61.

Roca-Dols A, Losa-Iglesias ME, Sánchez-Gómez R, Becerro-de-Bengoa-Vallejo R, López-López D, Rodríguez-Sanz D, et al. Effect of the cushioning running shoes in ground contact time of phases of gait. J Mech Behav Biomed Mater. 2018/08/25. 2018 Dec;88:196–200.

Methods are well-designed with relevant and complete information. Correct sample size calculations, good description of the properties of the outcome measurements as well as detailed statistical analyses were included. Tables, figures and redaction of the results are presented in a correct way providing a good presentation of the main finding of the study.

In line 76 to 82 in methods section I suggest authors must include a reference to Ethics requirements Helsinki declaration

Holt GR. Declaration of Helsinki—The World’s Document of Conscience and Responsibility. South Med J. 2014 Jul;107(7):407–407.

Discussion section may include future research studies secondary to the current findings of this study. Clinical considerations, limitations and overall discussion are well-presented, but future research may be useful in order to propose future research regarding this field.

Finally, I suggest to authors in discuss section to include the possbilty of influence of aging or even another pathologies like Parkinson diseease in your study finding suggest to include this references include in the attached to complete this requirement.

In line 198-214

Brognara, Navarro-Flores, Iachemet, Serra-Catalá, Cauli. Beneficial Effect of Foot Plantar Stimulation in Gait Parameters in Individuals with Parkinson’s Disease. Brain Sci. 2020 Jan 27;10(2):69.

6. PLOS authors have the option to publish the peer review history of their article (what does this mean?). If published, this will include your full peer review and any attached files.

Reviewer #1: No

Reviewer #2: No

---

## [Author Response · Author response to Decision Letter 0]

20 Apr 2020

First of all, we would like to thank the academic editor and reviewers for their insightful and constructive comments, which are very helpful in improving the quality of the manuscript. We have addressed all the comments raised in the revised manuscript. The point-by-point responses to the comments are provided below.

Academic editor:

ABSTRACT

1. In the abstract, the results of the condition without foam must be provided.

Response: Thank you for the comments. We have added the related results accordingly in the abstract (Line 43-44). 

2. Why do the results of correlations appear in the abstract and are not explained as objectives? This is also the case in the introduction. Correlations appear as hypotheses but are omitted as objectives. Nor is there any rationale in the introduction that explains the need to evaluate these relationships between variables. Please add it or remove the correlations analyses.

Response: Thank you for the comments. We have removed the correlation analyses in the abstract.

3. The conclusions should also be specific as to whether you are comparing with the firm ground or with foam.

Response: It is in the foam(unstable) surface. We have specified the comparison in the results accordingly (Line 46).

4. Add keywords.

Response: We have added key words as suggested (Line 48).

INTRODUCTION

5. After conducting a general review of the article, it is observed that the type of sole has no effect on the firm ground. It only has an effect when tested on an unstable surface. In this respect, the only thing the authors contribute is "As the healthy participants could achieve their best performance on a firm surface [7, 9], it may not be sufficiently discriminating to detect the change of postural sway across footwear conditions. Indeed, textured insole was reported to reduce the postural sway when standing on foam, when compared to the barefoot condition [9, 10].” The objective of the study should not be to force the support condition to observe differences (fishing), in postural sway, as stated in the paragraph, but rather to determine what is the need to evaluate on unstable surfaces (are they common in the population?) and from there see if differences exist or not according to the type of sole. The approach of the study should change so that the objectives of the study make sense and serve to contribute to the field of footwear design.

Response: Thank you very much for your insightful comments. We agree with your suggestion. Now we have made a substantial revision in the introduction part to address the uniqueness of our footwear insole design, and presented a strong justification to conduct the study (Line XX).

Methods

6. Please review methodological guidelines, such as CONSORT for example, to add the missing methodological aspects that allow replicability of the data. For example, you should add recruitment time, reliability of measurement tools (StimUp® Balance 5 99 Pad, Will Medical, Tokyo, Japan; Pedar®, Novel GmbH, Munich, Germany), etc.

Response: Thank you for your comments. We have added the information about participant recruitment (Line 100-103) and the reliability of measurement (Line 139-140).

Statistics

7. Only EMGmvc% was positively skewed? Indicate, if it is true, that the others met the assumption of normality. The statistics used for the main study are very well designed. However, here in this section, there is also no reference to correlations of muscle activation with postural sway.

Response: We have tested the data normality and we confirmed that only EMGmvc% was positively skewed. Additional information about the statistics analysis have been provided (Line 167-168). Per your suggestion, we have removed the correlation analysis in the present study. 

Results

8. Change the + symbols to parentheses.

Response: We have revised accordingly (Line 180). 

9. The results do not show the correlations that are in the abstract either.

Response: Sorry for the mistake but we have removed the correlation analysis in the study.

10. Change "indicates positive interaction effects" by "significant interaction"

Response: We have revised accordingly (Line 185).

11. Please, add the legends to the figures.

Response: The legends of Figure 1 is in Line 120-122, Figure 2 in Line 197-198, Figure 3 in Line 200-201.

Discussion

12. More information on the mechanisms by which alterations in postural control and muscle activation occur is lacking in the discussion. As well as discussing the differences between what happens on firm ground and on unstable surfaces.

Response: Thank you for your suggestion. We have provided an elaborated discussion about the mechanism of the insole design and the firm/foam ground in the discussion section (Line 212-237).

13. It should be added, in limitations, that the use of foam, as a form of unstable soil does not have much external validity as it is not representative of real life. It would be advisable to slightly modify what the authors have put forward.

Response: We have highlighted the limitation in the discussion accordingly (Line 258-259).

Reviewer #1: 

This paper examined the footwear insole texture and supporting surface conditions on static postural stability and lower limb muscle activation for health older people. While this is a potential very valuable study there is some work needed to get it to the standard for this journal. There were some typographical errors I have included some in specific comments. Please refer to the specific comments below:

1. The effect of support surface and footwear condition on postural sway and lower limb muscle action of the old people

It is suggested that the topic should be change to "The effect of support surface and footwear condition on postural sway and lower limb muscle action of the old women' because only women participants involved in this study.

Response: Thank you for the insightful comments. We have changed the title as suggested. 

2. Start with capital letter for 'compared' in the Results abstract section.

Response: Rectification has been made accordingly (Line 37).

3. Start with capital letter for 'for' in the Conclusions abstract section.

Response: Rectification has been made accordingly (Line 44).

4. Change 'old people' to 'old women' in the Background section.

Response: Rectifications has been made accordingly.

5. What correlation methods/analysis used for your study? There were correlation results in the Abstract but the information was not available in the Results section. There were also no Correlation Analysis information found in the Data Analysis section. The correlation results also were not discussed in the Discussions section. Please revise this issues accordingly.

Response: Thank you for your comments. We have removed the correlation analysis in the present study.

6. 'pedar' should start with capital letter i.e. line 130...Please check and revise accordingly.

Response: Rectification has been made accordingly (Line 137).

7. It is good to state the minimum age of your participants, i.e. how many participants are age from 60-65 years old. In the discussion section, you mentioned 'elderly' and by definition, elderly refers to person who are age 65 years old and above...but in Table 1, the age minimum age range of your participants are 60 years old..please revise according regarding the use of 'elderly' and older population/older women.

Response: Thank you for the comments. Now we used “old women” instead of “elderly” in the manuscript.

8. “To our knowledge, this is the first study that investigates the footwear condition and supporting surface conditions on static postural stability and lower limb muscle activation for healthy old women.”

“As to our knowledge, no studies have investigated the effect of footwear insole condition on both static postural stability and lower limb muscle activity for older people.”

Please standardized the term used for helthy old momen and older people. What is the main participants involved in this study?

Response: Thank you for the comments. We have removed that sentence. 

9. “As to our knowledge, no studies have investigated the effect of footwear insole condition on both static postural stability and lower limb muscle activity for older people.” The authors should provide a detailed argument before coming to this statement. The information in the same paragraph were only on textured insoles, spike insoles that was inserted to the participants shoes during the testing session. There should be an intensive information on how does the footwear used in this were different from the previous study. Is it readymade footwear that is available in the market?

Response: Thank you for your comments. We have added more information about our insole design in the introduction section. This is a footwear prototype that is yet available in the market (Line 75-83). 

10. How do you make sure that all participants have the minimum foot sensitivity level before they were accepted in your study? Do you conducted any test on foot sensitivity?

Response: Thank you for your expert comment. Yes, we did use Semmes Weinstein Monofilament test to evaluate the foot sensitivity for each participant. The information has been added to the method (Line 104-112) and result (Line 178-180) sections. 

Reviewer #2: 

1. I am grateful for the possibility to revise this research study. The effect of support surface and footwear condition on postural sway and lower limb muscle action of the old people is a trend topic in the current research literature and may be a main focus of interest for readers. 

Response: Thank you for the positive comments about our study.

2. Results of the abstract need to reflect the findings with respect to both groups and the lack of significant differences of balance, and also you need reflect the meaning of AP and ML because these abbreviates are not reflected clearly.

Response: We have modified the abstract accordingly (Line 40). 

3. Introduction may be improved adding new information in order to provide an adequate state-of-the-art including some references. I suggest to include this references include in the atteched to complet this requeriment

Lines 55-68

• Rodríguez-Sanz D, Tovaruela-Carrión N, López-López D, Palomo-López P, Romero-Morales C, Navarro-Flores E, et al. Foot disorders in the elderly: A mini-review. Disease-a-Month. 2018 Mar;64(3):64–91.

• Calvo-Lobo C, García AR, Iglesias MEL, López-López D, Rodríguez-Sanz D, Romero-Morales C, et al. The relationship between shoe fitting and foot health of persons with down syndrome: A case control study. Int J Environ Res Public Health. 2018 May 14;15(5).

• López-López D, Marañon-Medina J, Losa-Iglesias ME, Calvo-Lobo C, Rodríguez-Sanz D, Palomo-López P, et al. The influence of heel height related on quality of life on the foot in a sample of women. Rev Assoc Med Bras. 2018 Apr;64(4):324–9.

• Roca-Dols A, Elena Losa-Iglesias M, Sánchez-Gómez R, Becerro-de-Bengoa-Vallejo R, López-López D, Palomo-López P, et al. Electromyography activity of triceps surae and tibialis anterior muscles related to various sports shoes. J Mech Behav Biomed Mater. 2018/06/05. 2018 Oct;86:158–71.

• Roca-Dols A, Losa-Iglesias ME, Sánchez-Gómez R, López-López D, Becerro-de-Bengoa-Vallejo R, Calvo-Lobo C. Electromyography comparison of the effects of various footwear in the activity patterns of the peroneus longus and brevis muscles. J Mech Behav Biomed Mater. 2018/03/13. 2018 Jun;82:126–32.

• López-López D, Expósito-Casabella Y, Losa-Iglesias M, Bengoa-Vallejo RB de, Saleta-Canosa JL, Alonso-Tajes F. Impact of shoe size in a sample of elderly individuals. Rev Assoc Med Bras. 2016 Nov;62(8):789–94.

• López López D, Losa Iglesias ME, Becerro de Bengoa Vallejo R, Palomo López P, Morales Ponce Á, Soriano Medrano A, et al. Optimal choice of footwear in the elderly population. Geriatr Nurs. 2015/08/12. 2015;36(6):458–61.

• Roca-Dols A, Losa-Iglesias ME, Sánchez-Gómez R, Becerro-de-Bengoa-Vallejo R, López-López D, Rodríguez-Sanz D, et al. Effect of the cushioning running shoes in ground contact time of phases of gait. J Mech Behav Biomed Mater. 2018/08/25. 2018 Dec;88:196–200.

Response: Thank you for your suggested articles. We have cited the following references.

o Rodríguez-Sanz D, Tovaruela-Carrión N, López-López D, Palomo-López P, Romero-Morales C, Navarro-Flores E, et al. Foot disorders in the elderly: A mini-review. Disease-a-Month. 2018 Mar;64(3):64–91.

o López López D, Losa Iglesias ME, Becerro de Bengoa Vallejo R, Palomo López P, Morales Ponce Á, Soriano Medrano A, et al. Optimal choice of footwear in the elderly population. Geriatr Nurs. 2015/08/12. 2015;36(6):458–61.

4. Methods are well-designed with relevant and complete information. Correct sample size calculations, good description of the properties of the outcome measurements as well as detailed statistical analyses were included. Tables, figures and redaction of the results are presented in a correct way providing a good presentation of the main finding of the study. 

Response: Thanks for the positive comments.

5. In line 76 to 82 in methods section I suggest authors must include a reference to Ethics requirements Helsinki declaration

Holt GR. Declaration of Helsinki—The World’s Document of Conscience and Responsibility. South Med J. 2014 Jul;107(7):407–407.

Response: We have included this reference accordingly (Line 97).

6. Discussion section may include future research studies secondary to the current findings of this study. Clinical considerations, limitations and overall discussion are well-presented, but future research may be useful in order to propose future research regarding this field. 

Response: We have addressed potential research directions, e.g. involving different populations, in the discussion section (Line 249-250).

7. Finally, I suggest to authors in discuss section to include the possibility of influence of aging or even another pathologies like Parkinson diseease in your study finding suggest to include this references include in the attached to complete this requirement.

In line 198-214

Brognara, Navarro-Flores, Iachemet, Serra-Catalá, Cauli. Beneficial Effect of Foot Plantar Stimulation in Gait Parameters in Individuals with Parkinson’s Disease. Brain Sci. 2020 Jan 27;10(2):69.

Response: Thank you for sharing this recent publication. We have cited this paper accordingly.

---

## [Decision Letter · Decision Letter 1]

12 May 2020

PONE-D-20-00586R1

The effect of support surface and footwear condition on postural sway and lower limb muscle action of the older women

PLOS ONE

Dear Dr. Yick,

Thank you for submitting your manuscript to PLOS ONE. After careful consideration, we feel that it has merit but does not fully meet PLOS ONE’s publication criteria as it currently stands. Therefore, we invite you to submit a revised version of the manuscript that addresses the points raised during the review process.

We would appreciate receiving your revised manuscript by Jun 26 2020 11:59PM. To enhance the reproducibility of your results, we recommend that if applicable you deposit your laboratory protocols in protocols.io, where a protocol can be assigned its own identifier (DOI) such that it can be cited independently in the future. For instructions see: http://journals.plos.org/plosone/s/submission-guidelines#loc-laboratory-protocols

We look forward to receiving your revised manuscript.

Kind regards,

Pilar Serra-Añó

Academic Editor

PLOS ONE

Reviewers' comments:

Reviewer's Responses to Questions

**Comments to the Author**

1. If the authors have adequately addressed your comments raised in a previous round of review and you feel that this manuscript is now acceptable for publication, you may indicate that here to bypass the “Comments to the Author” section, enter your conflict of interest statement in the “Confidential to Editor” section, and submit your "Accept" recommendation.

Reviewer #1: All comments have been addressed

Reviewer #2: All comments have been addressed

2. Is the manuscript technically sound, and do the data support the conclusions?

Reviewer #1: Yes

Reviewer #2: Yes

3. Has the statistical analysis been performed appropriately and rigorously? 

Reviewer #1: Yes

Reviewer #2: Yes

4. Have the authors made all data underlying the findings in their manuscript fully available?

Reviewer #1: Yes

Reviewer #2: Yes

5. Is the manuscript presented in an intelligible fashion and written in standard English?

Reviewer #1: Yes

Reviewer #2: Yes

6. Review Comments to the Author

Reviewer #1: Thanks for the edited version and all authors did a good job in revising the manuscript. However there are some recommendations for this reviewed version.

1. There is repeated information on ethics approval on line 90 and 99. Please revise accordingly.

2. 23 participants involved in this study (line 180) but this number is not tally with the reported number of the dominant leg (line 184). Please revise accordingly.

Reviewer #2: This study supports novel information about The effect of support surface and footwear condition on postural sway and lower limb muscle action of the old people

This is an interesting aim with the quality of life scope.

Authors have adressed all the required modifications in a correct way.

The redaction is clear and concise with appropriated scientific terms.

The sample size calculation, structured tables and methodology are adequate and provide important contents.

Therefore, this study may support considerations about The effect of support surface and footwear condition on postural sway and lower limb muscle action of the old people

7. PLOS authors have the option to publish the peer review history of their article (what does this mean?). If published, this will include your full peer review and any attached files.

Reviewer #1: No

Reviewer #2: No

---

## [Author Response · Author response to Decision Letter 1]

14 May 2020

First of all, we would like to thank the academic editor and reviewers for their constructive comments, which are very helpful in improving the quality of the manuscript. We have addressed all the comments raised in the revised manuscript. The point-by-point responses to the comments are provided below.

Academic editor:

To enhance the reproducibility of your results, we recommend that if applicable you deposit your laboratory protocols in protocols.io, where a protocol can be assigned its own identifier (DOI) such that it can be cited independently in the future.

Response: Thank you for your recommendation. We have added laboratory protocols following the guidelines.

Reviewer #1: 

1. There is repeated information on ethics approval on line 90 and 99. Please revise accordingly.

Response: Thank you for the error. We have removed the repeated information as suggested. 

2. 23 participants involved in this study (line 180) but this number is not tally with the reported number of the dominant leg (line 184). Please revise accordingly.

Response: Rectification has been made accordingly (Line 182).

---

## [Editor Report · Decision Letter 2]

20 May 2020

The effect of support surface and footwear condition on postural sway and lower limb muscle action of the older women

PONE-D-20-00586R2

Dear Dr. Yick,

We are pleased to inform you that your manuscript has been judged scientifically suitable for publication and will be formally accepted for publication once it complies with all outstanding technical requirements.

With kind regards,

Pilar Serra-Añó

Academic Editor

PLOS ONE

---

## [Editor Report · Acceptance letter]

22 May 2020

PONE-D-20-00586R2 

The effect of support surface and footwear condition on postural sway and lower limb muscle action of the older women 

Dear Dr. Yick:

I am pleased to inform you that your manuscript has been deemed suitable for publication in PLOS ONE. Congratulations! Your manuscript is now with our production department. 

With kind regards,

on behalf of

Dr. Pilar Serra-Añó 

Academic Editor

PLOS ONE